# Low-Noise Amplifier for Deep-Brain Stimulation (DBS)

**Tiago Matheus Nordi** [1], **Rodrigo Henrique Gounella** [1], **Maximiliam Luppe** [1], **João Navarro Soares Junior** [1], **Erich Talamoni Fonoff** [2], **Eduardo Colombari** [3], **Murilo Araujo Romero** [1] and **João Paulo Pereira do Carmo** [1,*]

1. Group of Metamaterials Microwaves and Optics (GMeta), Department of Electrical Engineering (SEL), University of São Paulo (USP), Avenida Trabalhador São-Carlense, Nr. 400, Parque Industrial Arnold Schimdt, São Carlos 13566-590, SP, Brazil; tmnordi@usp.br (T.M.N.); rodrigogounella@usp.br (R.H.G.); maxluppe@sc.usp.br (M.L.); navarro@sc.usp.br (J.N.S.J.); murilo.romero@usp.br (M.A.R.)
2. Department of Neurology, Faculty of Medicine, Avenida Dr. Arnaldo, Nr. 455, Cerqueira César, São Paulo 01246-903, SP, Brazil; fonoffet@usp.br
3. Department of Physiology and Pathology, Faculty of Odonthology, São Paulo State University (UNESP), Rua Humaitá, Nr. 1680, Araraquara 14801-385, SP, Brazil; eduardo.colombari@unesp.br
* Correspondence: jcarmo@sc.usp.br

**Abstract:** Deep-brain stimulation (DBS) is an emerging research topic aiming to improve the quality of life of patients with brain diseases, and a great deal of effort has been focused on the development of implantable devices. This paper presents a low-noise amplifier (LNA) for the acquisition of biopotentials on DBS. This electronic module was designed in a low-voltage/low-power CMOS process, targeting implantable applications. The measurement results showed a gain of 38.6 dB and a −3 dB bandwidth of 2.3 kHz. The measurements also showed a power consumption of 2.8 μW. Simulations showed an input-referred noise of 6.2 μV$_{RMS}$. The LNA occupies a microdevice area of 122 μm × 283 μm, supporting its application in implanted systems.

**Keywords:** CMOS; deep-brain stimulation (DBS); low-noise amplifier; implantable devices

## 1. Introduction

Over the past decades, neuroscientists have been engaging the integrated circuit community to help them in the development of new tools for analyzing and understanding the brain. In this context, fundamental in vivo research on small animals has to be performed, which requires miniaturized instrumentation for long-term studies [1]. For several years, scientists have speculated that electroencephalographic (EEG) activity might provide the communication channel between brain and computer [2]. As the field has evolved, the demand for more functionally and miniaturization from the electronics community have risen. Since it is necessary to deal with low-amplitude biological signals, it is important to design amplifiers that make these signals compatible with devices such as ADCs for further analysis on computers. The amplifiers must have specific requirements, such as providing selective amplification to the physiological signal, rejecting superimposed noise and interference signals, and assuring protection from damage caused by high voltages and currents [3].

The recent developments of microelectronics have resulted in new applications involving the acquisition of biosignals both with wearable and implantable devices [4–8]. For instance, the electrocardiogram (ECG) is one of the most well-known applications, consisting of the acquisition of biosignals to allow medical doctors to diagnose heart diseases [6–10]. The electroencephalogram (EEG) is another widespread application with a large number of newly published works every year [11–13]. The neural recording has pushed the acquisition of biosignals to new levels, with new applications involving neuromodulation [14–16]. Such applications include optogenetics, which is an emergent field of applications, where the signals are acquired from a specific part of the brain, while simultaneously, this same region of the brain can also be stimulated with light [17–20].

Then, a new paradigm of optogenetics is the concept of an electrode with a chip-in-the-tip, where a bioamplifier and the respective signal processing/control/interface electronics can simultaneously acquire the neuronal signals while stimulating the brain with light [21,22].

Another emerging field of applications is the deep-brain stimulation (DBS) [23]. Deep-brain stimulation (DBS) involves implanting, through a surgical procedure, a medical device called a neurostimulator (often also called brain pacemaker). In the procedure, implantable semi-rigid tips (with electrodes at the ends) are also inserted at strategic points in the thalamus, subthalamic region, globus pallidus, among other areas [24]. The electrodes are then connected to the neurostimulator itself by means of extension cables containing metallic wires [25]. These electrodes are normally distributed at the end of the tips [26] which is inserted into the brain, steering them towards the desired neurostimulation zones. The neurostimulator sends mild pulses to the brain through the electrodes [27]. The electrical current used is very low and is injected into points in the brain which are mostly located in deep areas.

The neurostimulator is a device with dimensions no larger than a matchbox, with an attached battery to provide energy for operation [28]. Figure 1 shows the concept of DBS [29], where a bioamplifier (also known as low-noise amplifier (LNA)) and the respective signal processing/control/interface electronics simultaneously acquire the neuronal signals and provide stimulation. The neurostimulator is usually placed in the chest or in the abdominal area, under the skin, so that no parts are exposed or visible. The electrical stimulation modifies the functioning of the neurons around the tips, when the system is turned on, relieving the symptoms of various diseases.

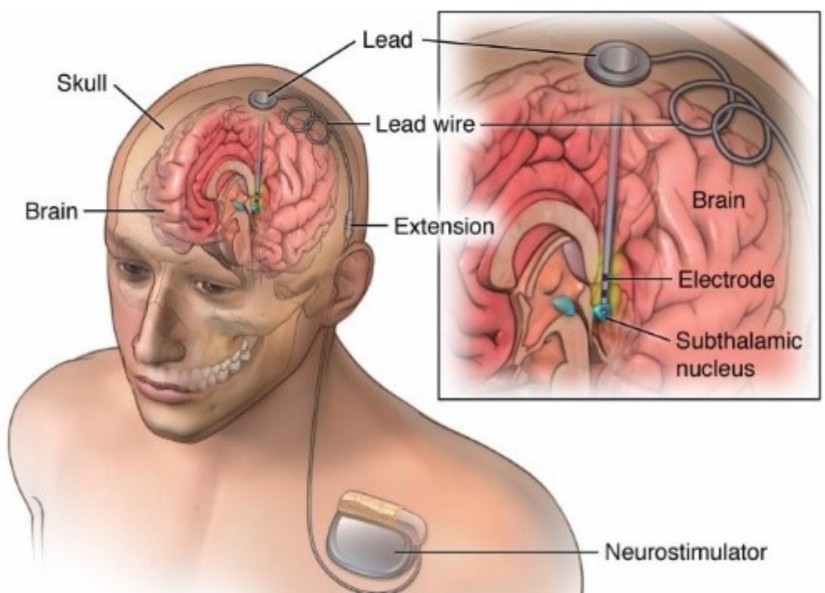

**Figure 1.** Concept of deep-brain stimulation (DBS).

The first current use of the DBS technique dates back to 1997, when it was authorized by the American FDA (Food and Drug Administration) for its application in the treatment of Parkinson's disease [30]. Since then, and thanks to their proven success, these systems have become first-line devices in therapies for the relief of symptoms associated with neurological and movement disorders that cannot be achieved with other therapies [31], e.g., chronic pain [32,33], Parkinson's disease [34,35], tremor [36,37], dystonia [38,39], morbid obesity [40], Tourette syndrome [41], essential tremor [42] and obsessive compulsive disorder [31].

Despite the great successes that were achieved, the neurostimulators are still quite large electronic devices that, in addition to using cables under the skin to connect to the head and then to the stimulation electrodes, still require the replacement of the power

battery via an invasive surgical procedure every two to four years. In the future, the implants must become more autonomous and less invasive, in order to reduce the heavy burden of replacing their batteries and the discomfort that the implant itself causes to the patient. There exists a pressure from the medical community and patient associations to reduce the discomfort caused by the implant, by reducing the size and weight, increasing the life of the implant through an efficient energy management system, and improving the operational safety, such as when performing magnetic resonance imaging (MRI) or even computed tomography (CT).

There are two paradigms for classifying deep-brain stimulation (DBS), the open-loop DBS (also known as conventional DBS) and the closed-loop DBS (closed-loop DBS also known as adaptive DBS) [43]. In the case of open-loop DBS, a neurologist manually adjusts stimulation parameters every 3–12 months after implantation. On the other hand, in the case of the closed-loop DBS, the adjustment of the stimulation parameters is performed automatically based on measured biomarkers. Biomarkers are acquired signals and can have different natures, namely bioelectrical, psychological, biochemical, among others [43]. Biomarkers are essential indicators on the disease under treatment with closed-loop DBS, because they help to adaptively reconfigure the signals used in neurostimulation [29]. The acquisition of biopotentials is an important component in closed-loop DBS.

For these reasons, and due to the lack in the market of miniaturized systems with potential for safe implantation, the design of CMOS microdevices comprising complete DBS systems has huge socioeconomic impacts both for make available new treatment techniques and to boost the market related to the area of medical instrumentation and healthcare. Figure 2 illustrates the block diagram containing the acquisition, neurostimulator and control modules of a CMOS microdevice for application on DBS. CMOS microdevices similar to the one presented in Figure 2 allow the integration of microsystems for DBS therapies, which have high potential for implantation in the brain. In this context, this paper presents the design of a fully integrated low-noise amplifier (LNA) suitable for recording biological signal within the range of sub-Hz up to 10 kHz.

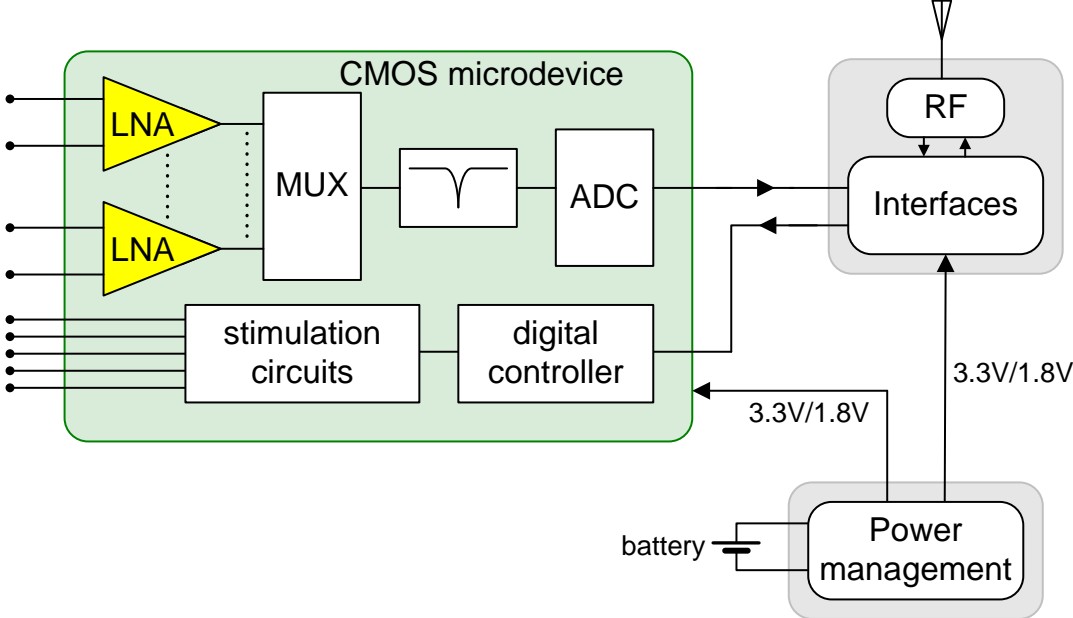

**Figure 2.** A block diagram of a CMOS microdevice containing the acquisition, neurostimulator and control modules for DBS. The LNA module presented in this paper is filled with the yellow color.

## 2. Design

A low-noise amplifier faces several challenges due to the nature of the signals to amplify, e.g., low-amplitudes and low-frequencies (and being very close to the DC component).

These types of amplifiers for neural recording typically present a mid-band gain of about 40 dB, and a bandwidth ranging from the sub-Hz to 10 kHz [44–53].

Figure 3a shows the schematic of the proposed LNA [46,47]. This amplifier is composed of the two pairs of capacitors $C_1$ = 20 pF and $C_2$ = 200 fF, a transconductance operational amplifier (OTA) and a pair of resistors $R_2$. The mid-band voltage gain of this amplifier is given by:

$$A_v = \frac{C_1}{C_2} = 100 = 40 \text{ dB} \tag{1}$$

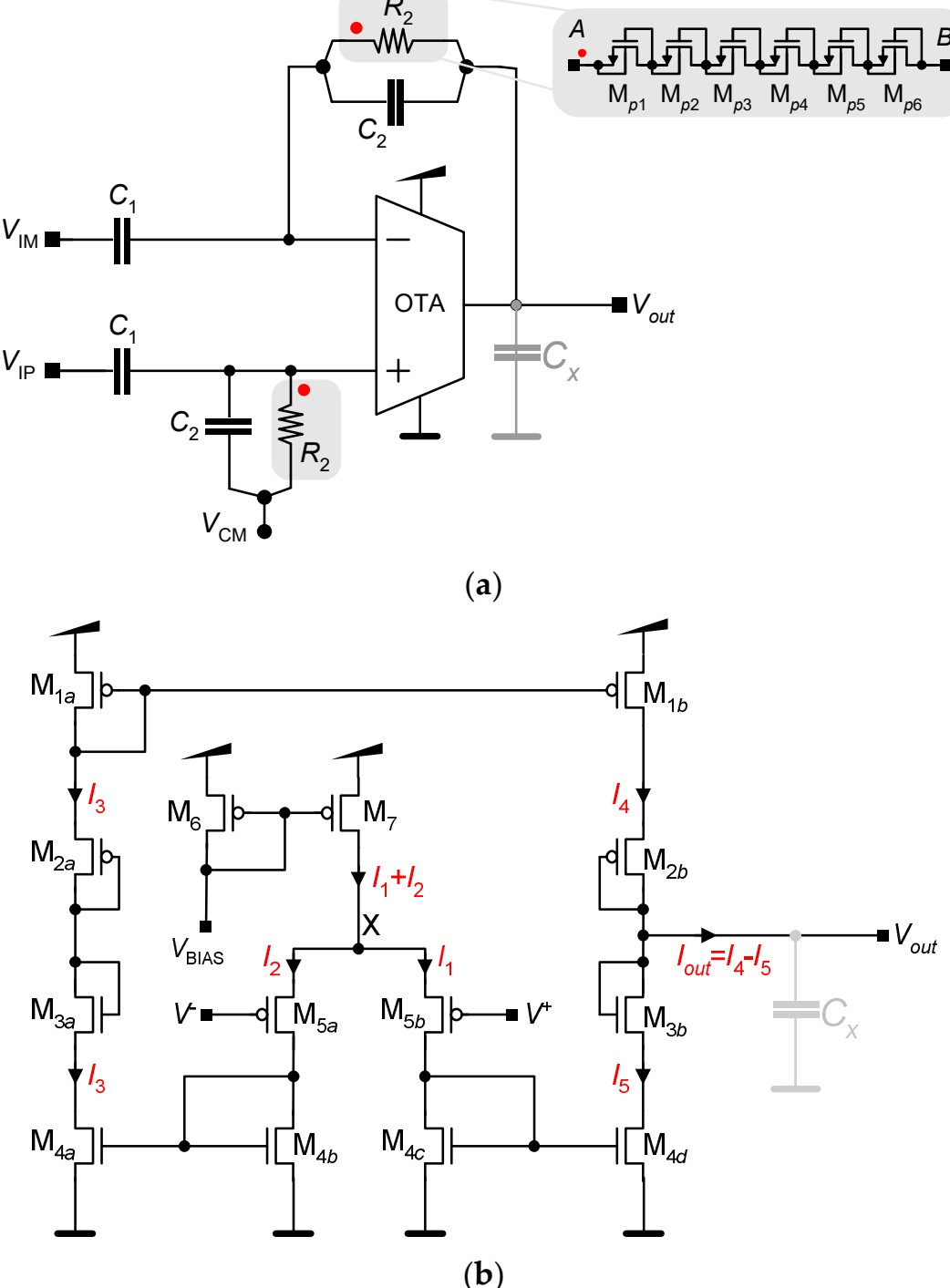

(**a**)

(**b**)

**Figure 3.** *Cont.*

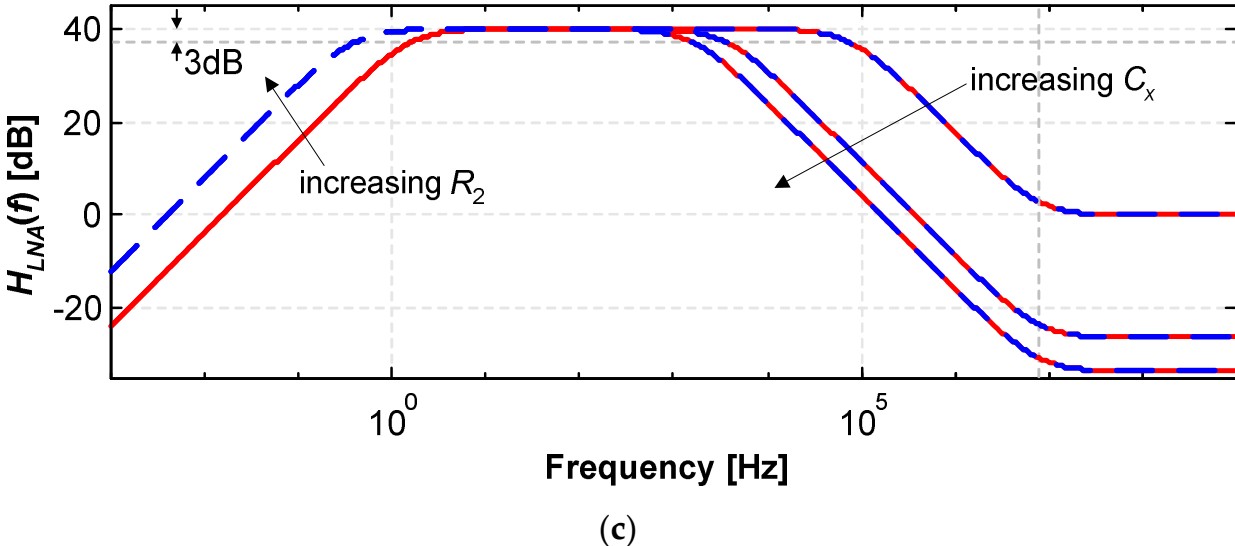

**(c)**

**Figure 3.** Schematics (**a**) of the low-noise amplifier, highlighting the pseudo-resistors components, and (**b**) of the OTA, showing the currents. (**c**) The transfer function of the LNA for six combinations of $R_2$ an $C_x$.

Figure 3b illustrates the schematic of the OTA, which is composed of eight PMOS and four NMOS: PMOS transistors $M_{5a}$ and $M_{5b}$ form a differential pair; NMOS transistors $M_{4b}$ and $M_{4c}$ forms the differential pair load and, also one to one current mirrors with $M_{4a}$ and $M_{4d}$, respectively; PMOS transistors $M_{1a}$ and $M_{1b}$ forms another one to one current mirror; PMOS transistor $M_6$ and $M_7$ form the biasing circuit; PMOS transistors $M_{2a}$ and $M_{2b}$ and NMOS transistors $M_{3a}$ and $M_{3b}$ are cascode transistors that increases the impedance of the nodes connected to their drains ($V_{out}$, for instance). The capacitor $C$ is only for understanding purposes because on the final design this one is replaced by $C_x$.

The OTA converts a voltage difference $V_d = V^+ - V^-$ into a current $I_{out}$. The conversion is such that the current at the output of the OTA is:

$$I_{out} = g_m \times (V^+ - V^-) \tag{2}$$

where $g_m$ is the transconductance of the OTA. The input signals $V^+$ and $V^-$ must have the same common-mode voltage $V_{CM}$ for a good working of the circuit, e.g., $V^+ = V_{CM} + v_d/2$ and $V^- = V_{CM} - v_d/2$. Under these circumstances:

$$I_{out} = g_m \times (V^+ - V^-) = g_m v_d \tag{3}$$

The transfer function $H_{LNA}(s)$ of the LNA in terms of the different components is given by (see Appendix A):

$$
\begin{aligned}
H_{LNA}(s) \quad &= \frac{V_{out}}{V_{IP} - V_{IM}} = -\left(\frac{C_1}{C_2}\right) \\
&\times \left[\left(\frac{C_2}{g_m}\right) \times \frac{\left(s\frac{C_2}{g_m} - 1\right) \times s}{s^2\left(\frac{C_1 C_2 + C_1 C_x + C_2 C_x}{g_m^2}\right) + s\frac{C_2}{g_m} + 1}\right]
\end{aligned}
\tag{4}
$$

where $g_m$ is the transconductance of the OTA, e.g., $g_m = I_{out}/(V^+ - V^-)$. The reason for the negative sign is explained in the demonstration in Appendix A. Appendix A presents the fully deduction of the transfer function $H_{LNA}(s)$ of the LNA. The capacitance $C_x$ includes all capacitances, either parasitic or connected, of the output node. The parasitic capacitances include the contributions of routing connections, the PADs for the exterior (when applicable)], the capacitance wires used for the measurements (when applicable), and the input capacitance of measurement instruments (when applicable).

The transfer function can be simplified in terms of the zeros and poles, resulting in:

$$H_{LNA}(s) = -\left(\frac{C_1}{C_2}\right) \times \left[\left(\frac{C_2^2}{C_1 C_2 + C_1 C_x + C_2 C_x}\right) \times \frac{\left(s - \frac{g_m}{C_2}\right) \times s}{(s + s_{p1}) \times (s + s_{p2})}\right] \qquad (5)$$

This transfer function can be rewritten as:

$$H_{LNA}(s) = -\left(\frac{C_1}{C_2}\right) \times \left[\left(\frac{C_2^2}{C_1 C_2 + C_1 C_x + C_2 C_x}\right) \times \frac{(s - 2\pi f_z) \times s}{(s + 2\pi f_L) \times (s + 2\pi f_H)}\right] \qquad (6)$$

The LNA transfer function contains two zeros, one zero located in the origin and another zero located at $f_z = g_m/(2\pi C_2)$, and two poles, one pole located in the lower cutoff frequency $f_L = 1/(2\pi R_2 C_2)$ and the other pole, in the upper cutoff frequency $f_H = g_m/[2\pi (C_1 + (C_1/C_2 + 1)C_X)]$. It must be noted that the frequency of the second zero $f_z$ is much higher than the frequency of any pole. Additionally, the LNA gain between $f_L$ and $f_H$ can be determined and its value is approximately $(C_1/C_2)$.

Figure 3b illustrates the transfer function $H_{LNA}(f)$ for six combinations of $R_2$ and $C_x$. The three plots represented in blue trace use a higher $R_2$ value and the three plots represented in dashed red traces use a lower value. Both the three blue and red plots were obtained with $C_x$ equal to 0 pF, 3.9 pF and 9.2 pF. Since the blue plots have a higher $R_2$, their lower cutoff frequency $f_L$ are smaller. The upper cutoff frequency $f_H$ decreases if the capacitance $C_x$ increases, maintaining constant the value of $R_2$. In conclusion, the bandwidth of the LNA increases if $R_2$ increases or $C_x$ decreases.

Figure 3c illustrates a Bode plot of a generic transfer function with two zeros, two poles, and their relative positions similar to the LNA zeros and poles of this work.

On a fully on-chip solution, the LNA connects internally to the next stage; thus, $C_x$ is only due the parasitic capacitance of internal connections. However, an internal capacitor $C_L$ or a switched capacitor array is included on many designs to trim the band, adjusting the pass-band to the desired application. This is achieved by changing the upper cutoff frequency $f_H$, trimming the capacitance $C_x$.

The resistor $R_2$ must present a very high value to guarantee a low cutoff frequency $f_L$, lower than 1 Hz. Since $C_2$ is in the order of few tenths of pF, $R_2$ must be in the order of TΩ. These resistors cannot be implemented in a conventional form in an integrated circuit; neither are commercially available, and if it was the case, the high tolerances would unbalance the circuit in Figure 3a with the two resistors away from each other by a few MΩ to a few GΩ. A widely known technique for implementation of high value resistors is the use of pseudo-resistors [46,53]. Figure 3a also details the implementation of resistors $R_2$ with pseudo-resistors. These pseudo-resistors are PMOS devices, as detailed with the zoom in the figure, each one composed of six PMOS transistors connected in series. It was found that these pseudo-resistors can reach values in the order of TΩ and occupy an area many orders of magnitudes lower than the area of a conventional resistor. They are called "pseudo" because it mimics the behavior of a real resistor. The red dots in the terminal A of the pseudo-resistors $R_2$ serves to show how these pseudo-resistors connects to the LNA. The terminal A of the pseudo-resistors connects to the bulk and source of $M_{p1}$, while the terminal B connects to the gate and drain of $M_{p6}$. The bulk of any PMOS $M_{p(i)}$ connects to the respective source, while the gate connects to the respective drain. Moreover, all PMOS are connected in series.

The most important characteristic of a LNA is its noise. The noise in our LNA is largely caused by the transistors of the OTA. The noise of MOS transistors can be modeled by two current sources from drain to source, and their power spectral density are given by:

$$i_{DSth}^2 \approx k_{th} kT g_m : \text{ thermal noise} \qquad (7)$$

$$i_{DSf}^2 \approx \frac{k_f}{C_{ox}WL}\frac{1}{f} : \text{ flicker noise} \tag{8}$$

where $k_{th}$ and $k_f$ are parameters that depend on the fabrication process, $k$ is the Boltsmann constant, $T$ is the temperature in Kelvin, $C_{ox}$ is the gate oxide capacitance, $W$ and $L$ are the transistor dimensions, and $g_m$ is the transistor transconductance.

To evaluate the effect of noise introduced by the transistors, the following procedure is performed:

(1)  Transpose the transistor noise current sources to the $V_{out}$ node;
(2)  Find the transfer functions between a current source $I_{out}$ applied to the output and $V_{out}$;
(3)  Find *PSD* of the noise at $V_{out}$;
(4)  Find the input referred noise.

The transposition of the noise current sources is easily performed since in the OTA the noise currents are mirrored to the output. Therefore, the total current at the output node is given by:

$$
\begin{aligned}
i_{outn}^2 \quad &\approx 2k_{th}kTg_{m5a} + 2\frac{k_f}{C_{ox}W_{5a}L_{5a}}\frac{1}{f} + 4k_{th}kTg_{m4c} + 4\frac{k_f}{C_{ox}W_{4c}L_{4c}}\frac{1}{f} \\
&+ 2k_{th}kTg_{m1a} + 2\frac{k_f}{C_{ox}W_{1a}L_{1a}}\frac{1}{f}
\end{aligned}
\tag{9}
$$

Notice that the noise of the cascode transistors does not affect the OTA noise.

The transfer function $H_{out}(s) = V_{out}(s)/I_{out(s)}$ should be deduced as done with $H_{LNA}(s)$. The final expression can be found and is presented below

$$H_{out}(s) = \frac{(R_2(C_1 + C_2)s + 1)}{s^2[R_2(C_1C_2 + C_1C_x + C_2C_x)] + s(C_1 + g_mR_2C_2 + C_x) + g_m} \tag{10}$$

The transfer function can be simplified, resulting in:

$$H_{out}(s) = \frac{(C_1 + C_2)}{(C_1C_2 + C_1C_x + C_2C_x)} \times \frac{\left(s + \frac{1}{R_2(C_1+C_2)}\right)}{(s + s_{p1}) \times (s + s_{p2})} \tag{11}$$

Function $H_{out}(s)$ has the same poles as $H_{LNA}(s)$ and a unique zero: $zero = 1/(R_2(C_2 + C_2))$, $s_{p1} = 1/(R_2C_2)$ and $s_{p2} = g_m/((C_1 + (C_1/C_2 + 1)C_x))$. The pass-band of LNA is located between $s_{p1}$ and $s_{p2}$. The *PSD* of the noise at the output now can be written, resulting in:

$$PSD_{Vout}(f) = \left| \frac{(C_1 + C_2)}{(C_1C_2 + C_1C_x + C_2C_x)} \frac{\left(s + \frac{1}{R_2(C_1+C_2)}\right)}{(s + s_{p1}) \times (s + s_{p2})} \right|^2 i_{outn}^2 \tag{12}$$

In the LNA pass-band, between $s_{p1}/2\pi$ and $s_{p2}/2\pi$, the *PSD* value is given by:

$$PSD_{Vout}(f) = \left| \frac{(C_1 + C_2)}{C_2} \times \frac{1}{g_m} \right|^2 i_{outn}^2 \tag{13}$$

These relation points out that it is important to keep $g_m$ high in order to reduce the output noise. This goal is reached by using large widths for $M_{5a}$ and $M_{5b}$. Finally, the input referred total noise is:

$$
\begin{aligned}
\text{Total Noise}_{input} \quad &= \frac{1}{LNA_{GAIN}}\sqrt[2]{\int_{-\infty}^{+\infty} PSD_{Vout}(f)df} \\
&\approx \frac{C_2}{C_1}\sqrt[2]{\int_{-\infty}^{+\infty} PSD_{Vout}(f)df}
\end{aligned}
\tag{14}
$$

where $LNA_{GAIN}$ is the gain of the LNA.

Table 1 lists the dimensions of the MOSFETs that comprises the OTA and the pseudo-resistors. The layout issues will be further addressed on Section 3.1. The listed relations $(W/L)$ re-

fer to the total value. For example, the transistors $M_{1a}$ and $M_{1b}$ with $(W/L)_1$ = (13.4 μm/20 μm) are composed of two parallel transistors, whose dimensions are equal to $(W/L)$ = (6.7 μm/20 μm) and at the same time, containing only one finger. In another example, e.g., for the transistors $M_{5a}$ and $M_{5b}$ with $(W/L)_1$ = (463 μm/0.51 μm) are composed of two parallel transistors, whose dimensions are equal to $(W/L)$ = (231.5 μm/0.51 μm) and, at the same time, containing 50 fingers for each parallel transistor with $(W/L)_{\text{finger}}$ = (4.63 μm/0.51 μm).

**Table 1.** Dimensions of the MOSFETs that comprises the OTA and the pseudo resistors.

| MOSFET | Total (*W/L*) | Multiplier (Parallel MOSFETs) | Fingers/Multiplier |
|---|---|---|---|
| $M_{1a}$, $M_{1b}$ | 13.4 μm/20 μm | 2 | 1 |
| $M_{2a}$, $M_{2b}$ | 20.6 μm/0.28 μm | 2 | 1 |
| $M_{3a}$, $M_{3b}$ | 15.4 μm/0.28 μm | 2 | 1 |
| $M_{4a}$, $M_{4b}$, $M_{4c}$, $M_{4d}$ | 10 μm/20 μm | 2 | 1 |
| $M_{5a}$, $M_{5b}$ | 463 μm/0.51 μm | 2 | 50 |
| $M_6$, $M_7$ | 2.3 μm/5.1 μm | 1 | 1 |
| Pseudo-resistors $M_{p1}$ to $M_{p6}$ | 1 μm/1 μm | 1 | 1 |

## 3. Implementation and Simulations

### 3.1. Layout Issues

Figure 4a illustrates the modifications made to schematics of LNA of Figure 3a for the fabrication. Each output node has a resistor of small value ($\approx$497.6 Ω) as a preventive protection against connection mistakes such as accidental short-circuits, limiting the output current. Each node with input signals, nodes with reference voltage and biasing nodes has a protection against electrostatic discharges (ESD) [54], an additional resistor $R_{in}$ of small value ($\approx$497.6 Ω) and a NMOS. Figure 4b illustrates the schematics of both the ESD protection (on left) and the resistor with NMOS (on right). This last resistance provides an additional level of protection to the gates of the internal circuits. The NMOS presents a width of 3 μm and a length of 1 μm. The resistance $R_{\text{BIAS}}$ in the biasing pin is also equal to a small value ($\approx$497.6 Ω). The capacitance $C_{\text{BIAS}}$ on bias voltage reduces the noise to provide the most stable bias voltage $V_{\text{BIAS}}$ as possible. This capacitance comprises three MIM capacitors with 2 fF/μm$^2$, each one with a total capacitance of 456 fF each.

The layout of ESD protections is similar to those proposed by Baker on chapter 4 of his book [55], which is composed of N$^+$/P-sub and P$^+$/N-well diodes. Figure 4c illustrates the layout of the ESD protections side-by-side with the respective photograph that was integrated in the fabricated CMOS microdevice. Each N$^+$/P-sub and P$^+$/N-well diodes are composed of the parallel of two smaller diodes measuring 16 μm × 87 μm. It must be noted that each PAD occupies an area of 62 μm × 62 μm.

The MOSFETs $M_1$ to $M_5$ of the OTA in the LNA were drawn with the technique known as common centroid, which provides circuits more resilient to process variations by matching the characteristics of the transistors [55,56]. Each MOSFET was split in two to make these devices immune from cross-chip gradients. Moreover, the gates of the MOSFETs $M_5$ (where the inverting and non-inverting input are connected) were split in several fingers to provide the best matching performance possible and reduce the parasitic capacitances, which are extremely high because of their widths [52]. Figure 5a illustrates the layout of the complete LNA side by side with the respective photograph. Figure 5b illustrated a zoomed view of the layout of the LNA without the array of capacitors $C_1$ for a better visualization. The tags (1) to (3) refers to the sets of input resistances $R_{in}$ with NMOS in Figure 4a. The capacitors $C_{\text{BIAS1}}$, $C_{\text{BIAS2}}$ and $C_{\text{BIAS3}}$ refers to the three MIM capacitor with 456 fF. Each one of the four capacitors $C_{2a}$ and $C_{2b}$ were also implemented with MIM capacitors with 100 fF. The common centroids of $M_1$ to $M_5$ of the OTA can be observed.

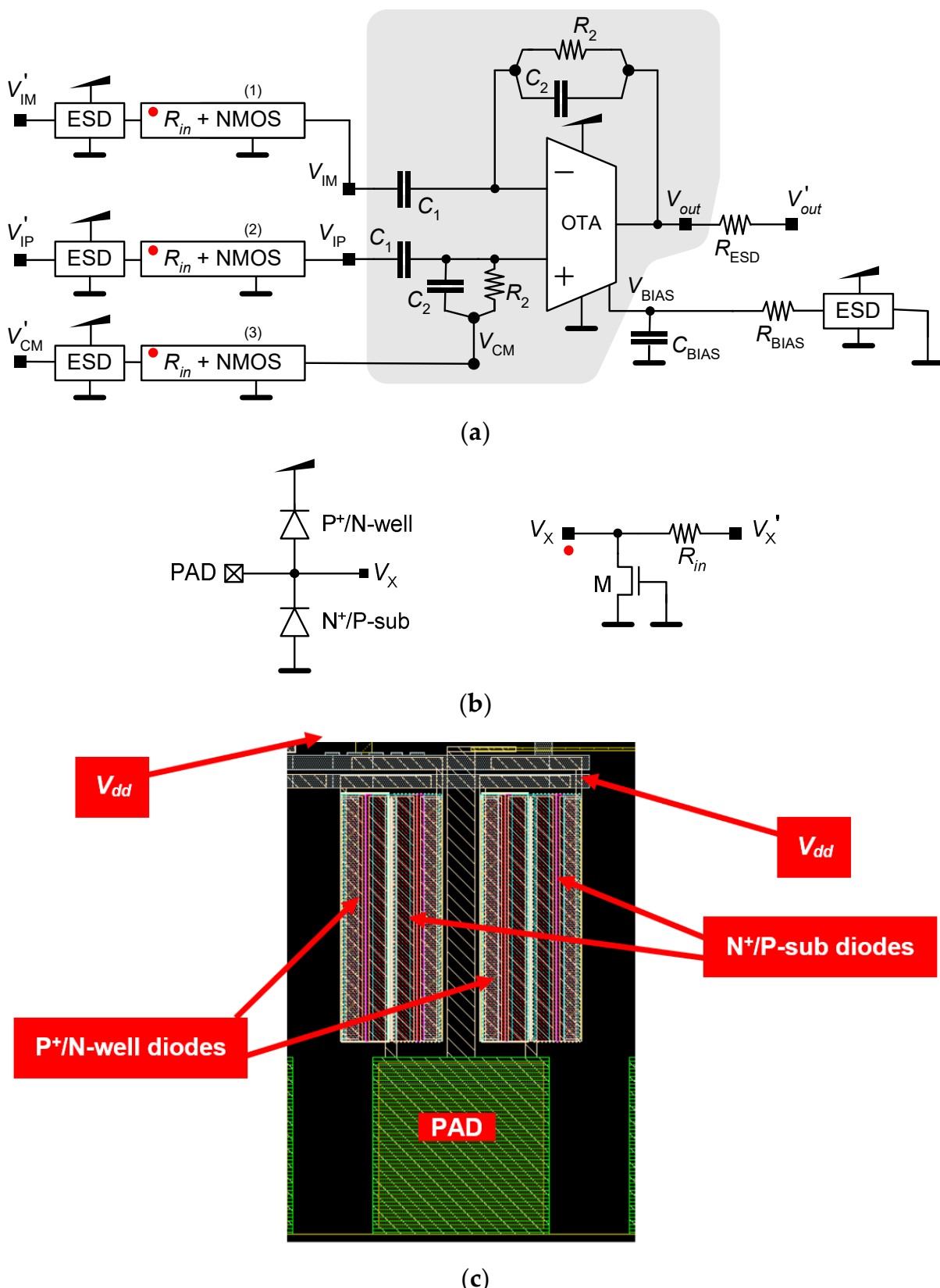

**Figure 4.** (**a**) Illustration of how the ESD protections, input resistance $R_{in}$ with NMOS and output resistances $R_{ESD}$ are connected to the low-noise amplifier. (**b**) Schematics of ESD protections (on left) and resistance with NMOS (on right). (**c**) Layout of the ESD protections [56].

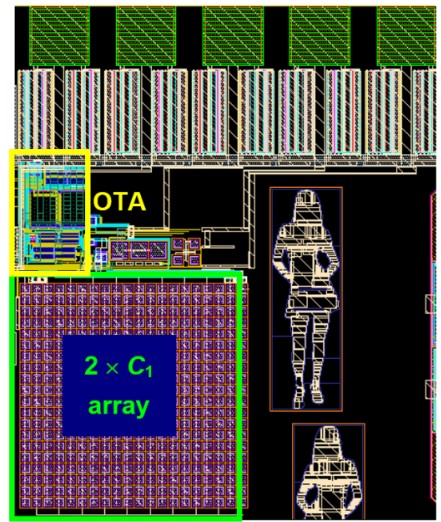
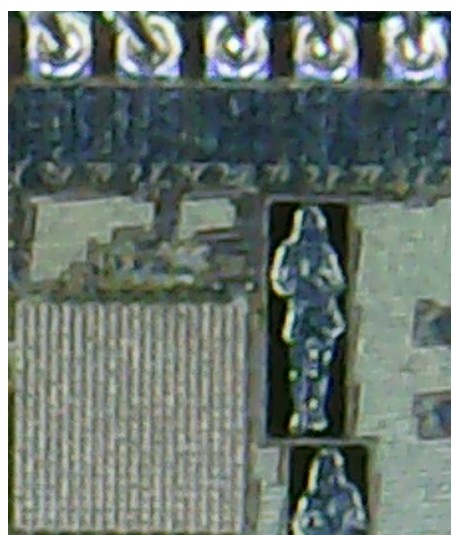

(**a**)

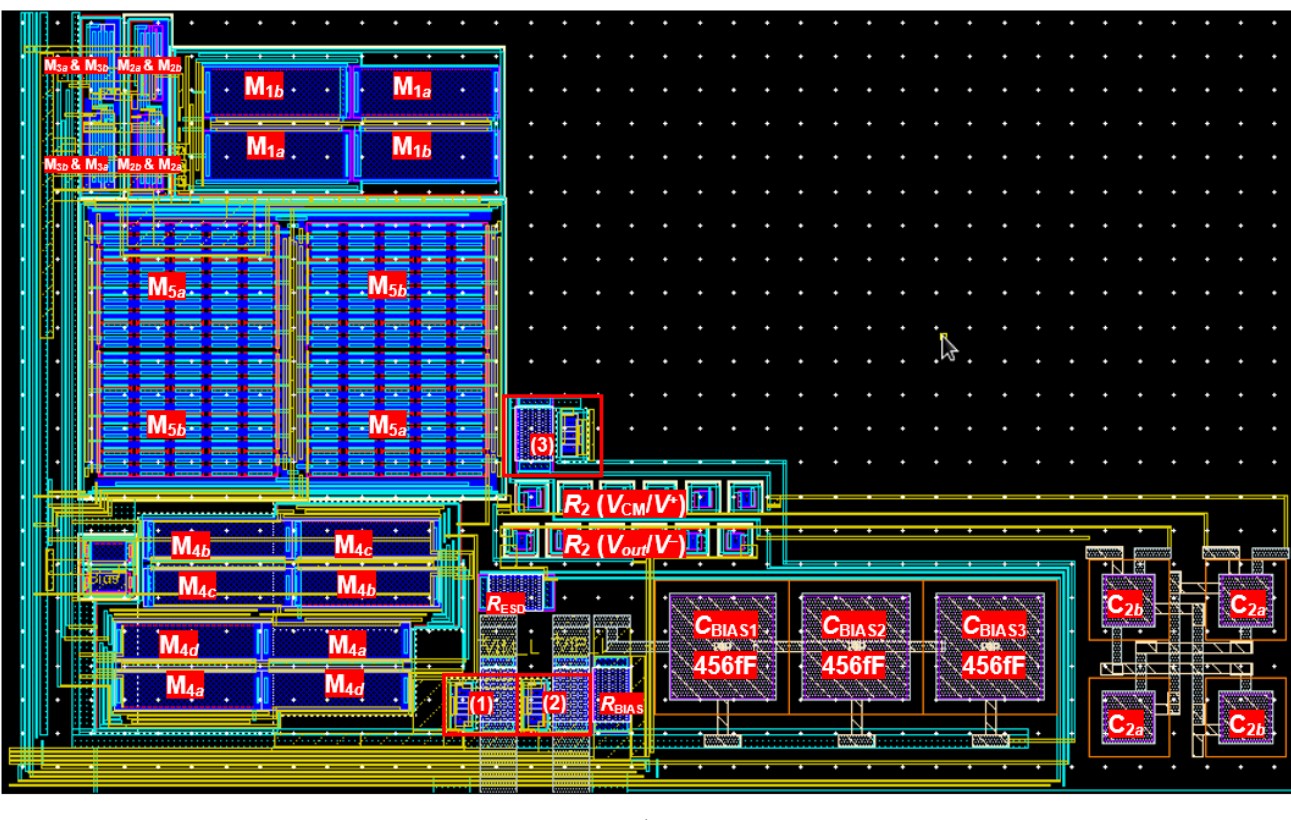

(**b**)

**Figure 5.** (**a**) Layout of the complete LNA side by side with the respective photograph, with a reference to the OTA and to the array of two capacitors $C_1$. (**b**) Zoomed view of the layout of the LNA without the array of capacitors $C_1$ for a better visualization.

### 3.2. Low-Noise Amplifier (LNA) Simulations

The behavior of the OTA was simulated in the H-Spice, and it was found a transconductance $g_{m5} \approx 7.58$ μS for the transistors $M_{5a}$ and $M_{5b}$. Thus, the transconductance of the OTA is also $g_m \approx 7.58$ μS, because it is the same value of $g_{m5}$.

The simulated LNA was the complete schematic of Figure 4a, taking into account the individual contributions of the ESD protections, the input resistor $R_{in}$ with NMOS and the output resistor $R_{ESD}$. The complete schematic was simulated to obtain a better preview

of the real conditions. Moreover, simulations were performed without the capacitance $C_x$ (e.g., $C_x = 0$) and with the passive voltage probe to understand the testing conditions. The oscilloscope used in the measurements was the Tektronix model MDO34 3-BW-100. It was used the passive voltage probe Tektronix model TPP0250, with an input capacitance of 3.9 pF.

Figure 6 shows the simulated resistance response of the pseudo-resistors in terms of the voltage $\Delta V = V_{in} - V_{out}$ at its terminals, where $V_{in}$ is the terminal that connects to the bulk of the first PMOS and $V_{out}$ is the terminal that connects to the gate of last PMOS (in concordance with Figure 3a). A voltage pulse source was placed between the $V_{in}$ and $V_{out}$ terminals of the pseudo-resistor, and the voltage were varied between $-0.4$ V and $0.4$ V. The current was simulated obtained, and the resistance was calculated by dividing the voltage by the current. Figure 6 shows the result of this simulation with the illustration of the voltage dependence of the pseudo resistance value.

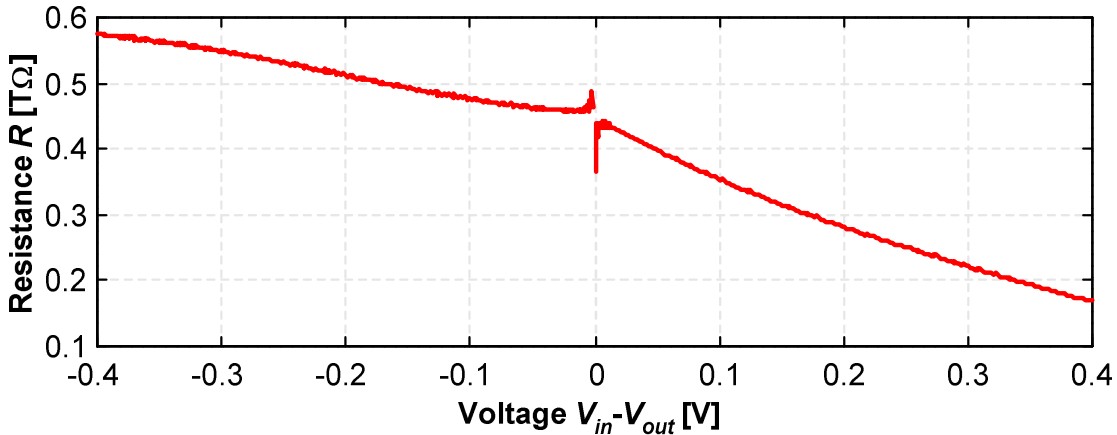

**Figure 6.** Voltage dependence of the pseudo resistance.

The total capacitance seen by the LNA output is the sum of the contributions of the parasitic capacitances of the internal metals connections, the PAD for wirebonding to the package, the connections of the test-bed, the 3.9 pF of the passive voltage probe and the cables for connect into the oscilloscope. The capacitance seen by the output of LNA was measured and determined to be equal to 5.3 pF, resulting in 9.2 pF total capacitance, if the voltage probe capacitance is also taken in account.

Figure 7 shows the gain simulation for $C_x = 0$, for $C_x$ equal to the capacitance of the voltage probe ($C_x = 3.9$ pF) and for $C_x$ equal to the total capacitance seen by the LNA output ($C_x = 9.2$ pF) to better understand the effect of the measurement setup.

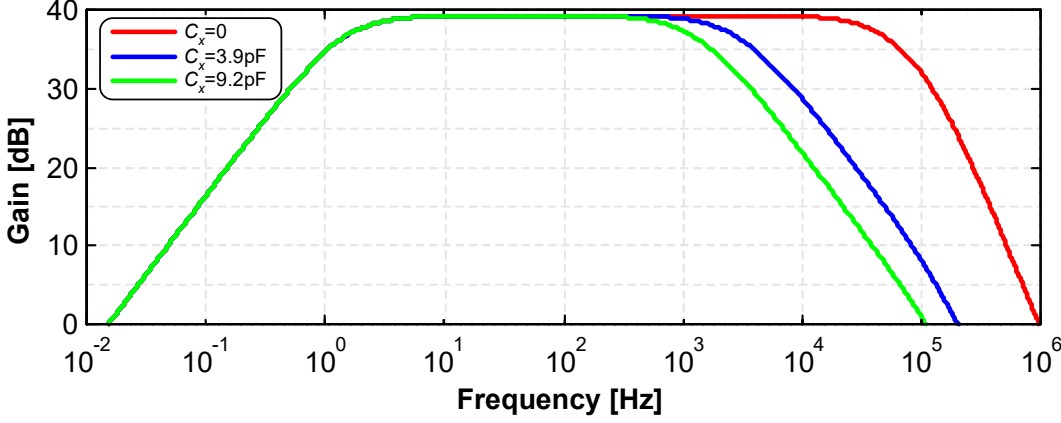

**Figure 7.** Simulated frequency response of the LNA for $C_x = 0$ (**red trace**), $C_x = 3.9$ pF (**blue trace**), and $C_x = 9.2$ pF (**green trace**).

The simulations showed that ideally with $C_x = 0$ the LNA presents a mid-band gain of $\approx$39.4 dB with a $-3$ dB bandwidth of $\approx$54 kHz. The simulations also showed that with the effect of the voltage probe, $C_x = 3.9$ pF, the LNA also presents a mid-band gain of $\approx$39.4 dB, but a $-3$ dB bandwidth of $\approx$3.1 kHz. For the case of $C_x = 9.2$ pF with the effect of total capacitance seen by the LNA output, the simulations showed a mid-band gain of $\approx$39.4 dB, but with a $-3$ dB bandwidth of $\approx$1.4 kHz.

Figure 7 shows the frequency response for frequencies higher that 0.01 Hz. The simulations revealed that this amplifier covers the range of extracellular recorded spikes, from 100 Hz to 6 kHz with a mid-band gain of $\approx$39.4 dB for the three situations of $C_x = 3.9$ pF.

As illustrated in Figure 8, two different scenarios were supposed, in order to simulate the robustness of the LNA considering the capacitance and resistance associate to the wires that connect the electrodes to the input of the LNA. A sinusoidal input with amplitude of 100 μV and a frequency of 1 kHz was considered. Moreover, the effect of $C_x$ was not considered because it makes no difference in the conclusions. The resistance was considered around 10 Ω in the situation 1. This value is probably higher than those found in a real situation with cables of good quality, but it was an extrapolated value to confirm the previous robustness of the amplifier. The situation 1 considered a serial resistance existent between the positive electrode $E^+$ and the positive input $V_{\mathrm{IP}}$ of the bioamplifier, and the one between $E^-$ and $V_{\mathrm{IM}}$. The situation 2 considered a capacitor of 100 nF placed in parallel to the voltage source. This situation is merely theoretical because this value is probably higher than those found in a real situation with cables of good quality, but once again, it is also a good test to the reliability of the amplifier. On both situations, the simulations showed an output signal with amplitude of 18.5 mV$_{\mathrm{PP}}$, e.g., a gain of 39.3 dB.

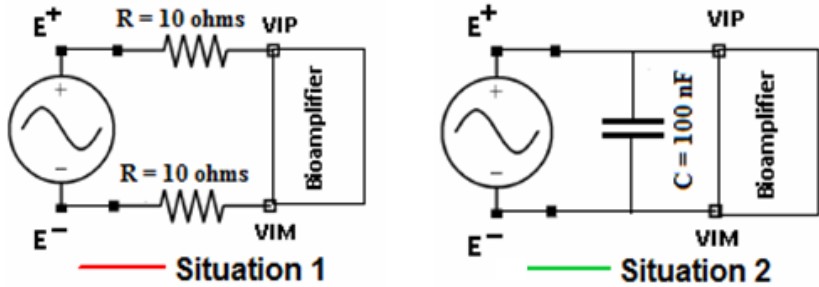

**Figure 8.** Scenarios for the simulation of the robustness of the LNA.

The simulated *PSD* of the output noise is presented in Figure 9. The $C_x$ value considered is 9.2 pF and two curves are traced, shown by a red and a blue curve. In the red curve, only the noise of the OTA transistors is taken in account; in the blue curve, the noise of the pseudo-resistors is also taken in account. The red curve behavior is exactly as described by expression (12). When the pseudo-resistors noise is added, the low frequency noise is increased.

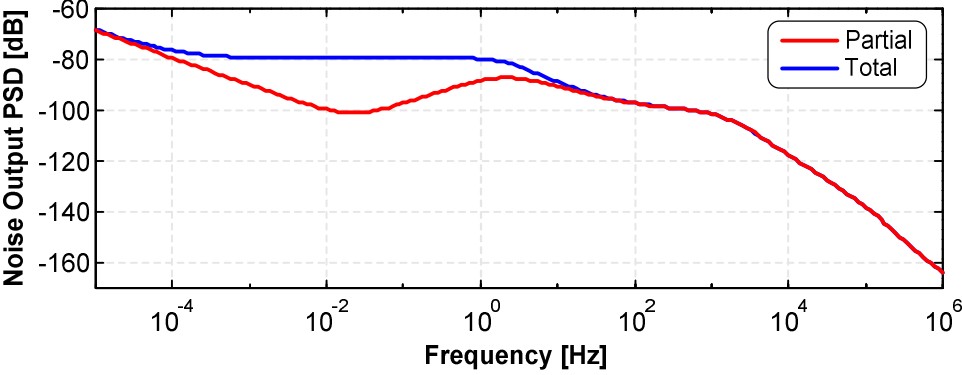

**Figure 9.** *PSD* noise of the LNA: in the red curve, only the noise of the OTA transistors is taken in account; in the blue curve, the noise of the pseudo-resistors is also taken in account.

The input referred noise of the LNA, find in the simulation, is 6.2 $\mu V_{RMS}$ (from 0.5 Hz to 50 kHz), when all noise sources are taken in account.

## 4. Experimental

### 4.1. Instruments and Setup

Figure 10 shows the schematic of the setup used during the measurements. This setup is composed of the microdevice under test itself, a test-bed board (shaded in gray) especially designed for the tests, an arbitrary signal generator Tektronix model AFG1022 with two simultaneous outputs and 25 MHz of bandwidth, an oscilloscope Tektronix model MDO34 3-BW-100 with four input channels and 100 MHz of bandwidth, passive voltage probes Tektronix model TPP0250, an external protoboard to facilitate the connection of bias resistors, and a multimeter to measure the voltage supply to ensure that it is within the valid tolerance range and/or other signals such as references and common mode voltages. The photograph shows a specific situation of testing. The external connections can be maintained unaltered to test other blocks in the microdevice, simply by redirecting the signals throughout dip-switches. Moreover, the dip-switches also can activate and deactivate several blocks within the microdevice. The test-bed board was designed to be supplied by a DC power jack, targeted to a typical supply voltage of 5 V. The test-bed can support supply voltages up to 16 V, whose value is limited by the voltage regulator TLV1117. This voltage regulator provides the required nominal voltage of 1.8 V to supply the CMOS microdevice. The common mode voltage $V_{CM}$ required to make the LNA work can be achieved in two ways: from an external voltage source or from an operational amplifier LM358, placed in the test-bed itself, working as voltage follower of half the nominal supply voltage, 0.9 V. A dip-switch allows the selection of the $V_{CM}$ source.

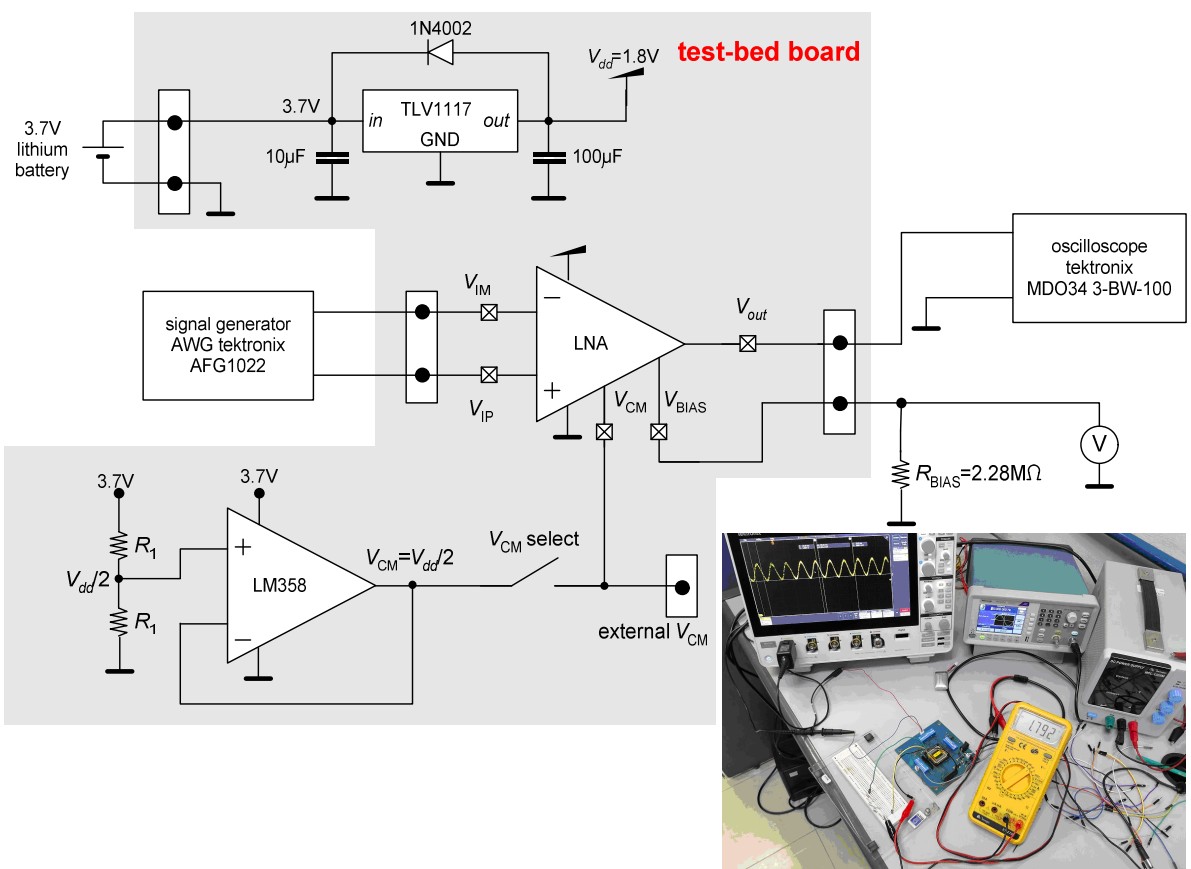

**Figure 10.** Schematic of the experimental setup for the characterization of the microdevice. The inset shows the setup photograph.

### 4.2. Results

Figure 11 shows the measured gain of the LNA and the simulated values. The amplitude of the input signals was settled to 4 mV$_{pp}$. The LNA presents a mid-band gain of $\approx$38.6 dB, which is close to the simulated results. Moreover, the $-3$ dB bandwidth was $\approx$2.3 kHz. For frequencies higher than 10 kHz, the measured gain approaches asymptotically to the simulated gain with $C_x = 9.2$ pF, although, for frequencies between 1 kHz and 10 kHz, the measured gain is slightly higher than the simulated gain.

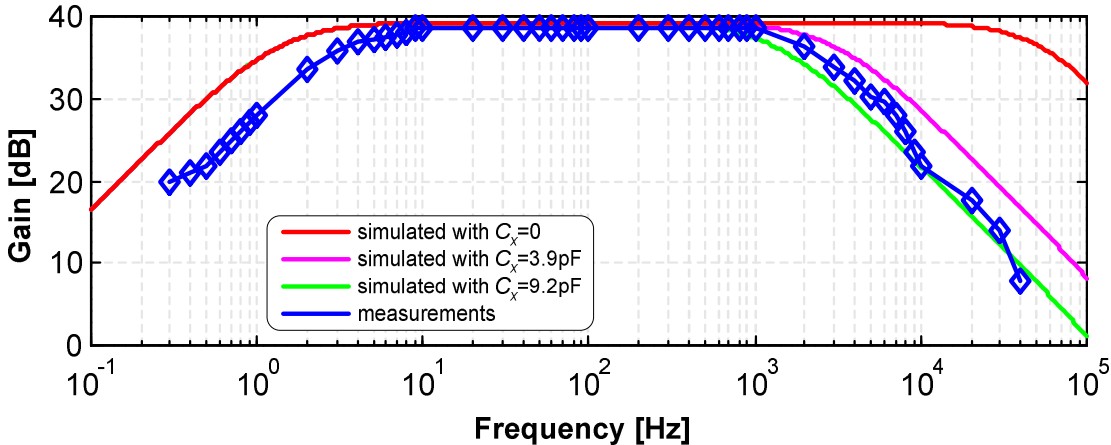

**Figure 11.** Measured gain and comparison with simulations for $C_x = 0$, $C_x = 3.9$ pF and $C_x = 9.2$ pF.

Notice that the upper cutoff frequency of the measured results is smaller than the ideally simulated value (pink plot in Figure 11, where $C_x = 0$ pF). It is caused by the contributions of the parasitic capacitances seen from the LNA output. The length of the cables to connect the test-bed to the oscilloscope was the smallest possible to decrease their contribution to the total capacitance $C_x$. The gain at 6 kHz is $\approx$29.5 dB, 9 dB below the mid-band gain, and is still an acceptable value. Higher gains are expected in a definitive LNA application, where the output of the LNA is directly connects to a multiplexer, and most of the parasitic capacitances are not present anymore. In other words, the total capacitance $C_x$ seen by the output of LNA will be drastically reduced, and therefore, the desired gain at 6 kHz will be increased.

An important test implemented is the characterization of the behavior of the gain to variations of the common-mode voltage $V_{CM}$. Figure 12 shows the measured gain for a common-voltage variation of $\pm0.1$ V and $\pm0.2$ V from the nominal value $V_{CM} = V_{dd}/2 = 0.9$ V. It is possible to observe that the gain has no sensitivity to small variations of common-mode voltage; thus, the LNA shows to be robust to these variations.

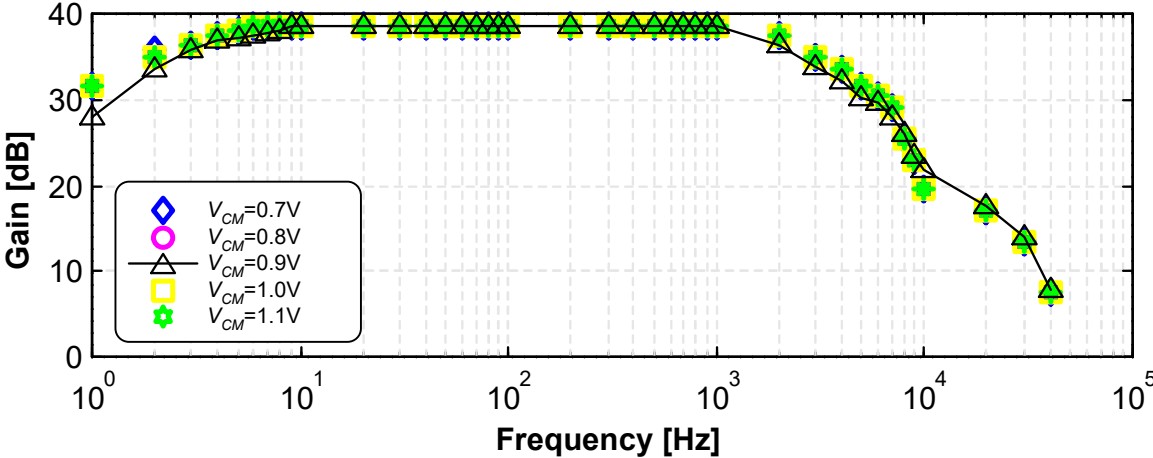

**Figure 12.** Measured gain in terms of the common-mode voltage $V_{CM}$.

A second set of stress tests performed with the LNA consisted of the injection of input signals with amplitudes capable to almost saturate the output either at 0 V or $V_{dd}$. Figure 13 illustrates the measurement results for these tests. The first input signals present an amplitude of 10 mV$_{pp}$ and interestingly, the gain was slightest higher than the gain obtained with an amplitude of 4 mV$_{pp}$, with a mid-band gain of ≈39.3 dB. The output signal in this first test presented a signal excursion of $\Delta V_{out}$ = 920 mV$_{pp}$, e.g., $\Delta V_{out}$ is almost equal to $V_{dd}/2$. The second test was more stressful, with an input amplitude of 20 mV$_{pp}$. In general, the gain is lower than those obtained with input signals with lowest amplitudes. This was almost expected because this pushes the output signal to present an excursion equal or higher than the supply voltage $V_{dd}$.

The two scenarios illustrated in Figure 8 were also tested. The amplitude of the input signals was settled to 4 mV$_{pp}$. These two scenarios are exaggerated when compared with real situations, but, for this reason, they are good for validating the robustness of the LNA. The results in Figure 13 revealed that the measured mid-band gain was almost equal to the values presented in Figure 11, where the measurement conditions were optimized with cables of high quality and short length.

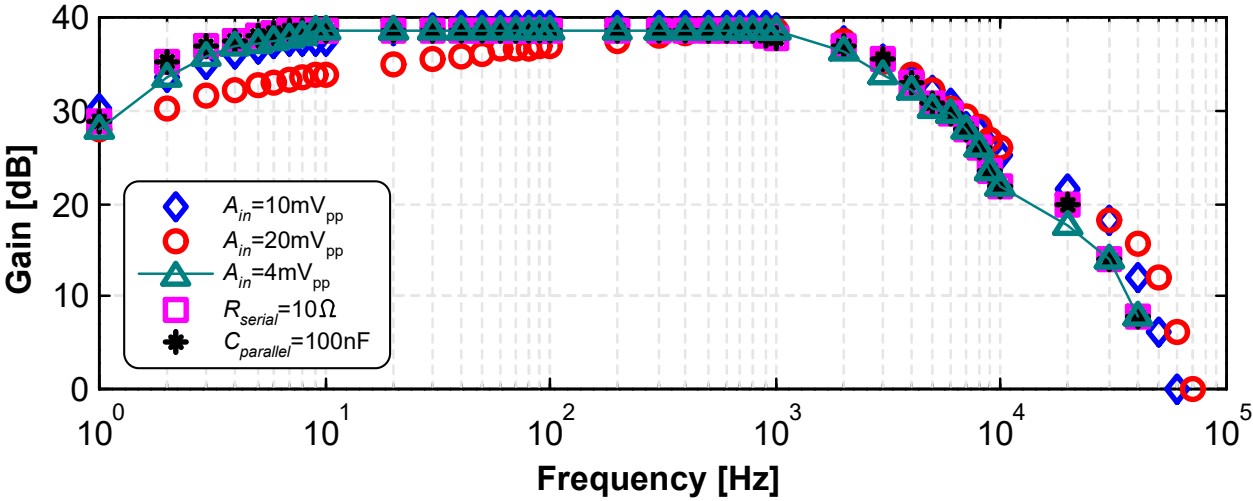

**Figure 13.** Measured gain for two sets of stress tests: input amplitudes of 10 mVpp and 20 mVpp, and cables from the electrodes with two serial resistance of 10 Ω and with a parallel capacitance of 100 nF.

The LNA was also tested with low-amplitude signals. These tests used a custom home-made signal generator able to generate sine waves with amplitudes of either 60 μV$_{pp}$ or 130 μV$_{pp}$. The behavior was not very different from those observed in Figure 11 with an amplitude of 4 mV$_{pp}$; however, the measured gain was slightly lower, e.g., ≈37.7 dB or less than 1 dB in relation to the 38.6 dB measured with the former.

A new set of tests were performed, each consisting of applying signals in saline solution with characteristics equivalent to those observed in neuronal tissues, to test the robustness of the LNA. Moreover, these saline tests were also performed to avoid ethical issues related to experimentation with in vivo human subjects and animals, and, at the same time, to get an idea about the phenomena in the brain.

In these tests, several electrodes were immersed in a jar filled with saline solution (saline solution consisting of sodium chloride solute dissolved in distilled water solvent in the proportion of 0.9%). The saline solution emulates very well the ionic species of the human tissue in terms of the electrical parameters.

Figure 14a shows the schematic of the experimental setup for these tests, which is composed of the signal generator, oscilloscope, test-bed board, CMOS microdevice and the bias resistor previously described. The power supply management is not displayed. Moreover, this setup is composed of a jar filled with a saline solution and by a tip with an array of electrodes. The tip is fabricated by additive manufacturing with 3D printing of PLA (polylactic acid) and filaments

with a diameter of 1.75 mm. The array of electrodes comprises a pair of injection electrodes and a pair of reading electrodes. It was performed a frequency sweep applied to a sinusoidal wave injected in the two injection electrodes. Then, the signals were sensed from the saline solution with the reading electrodes and further amplified by the LNA. Figure 14b illustrates a photograph of the tip with the array of electrodes.

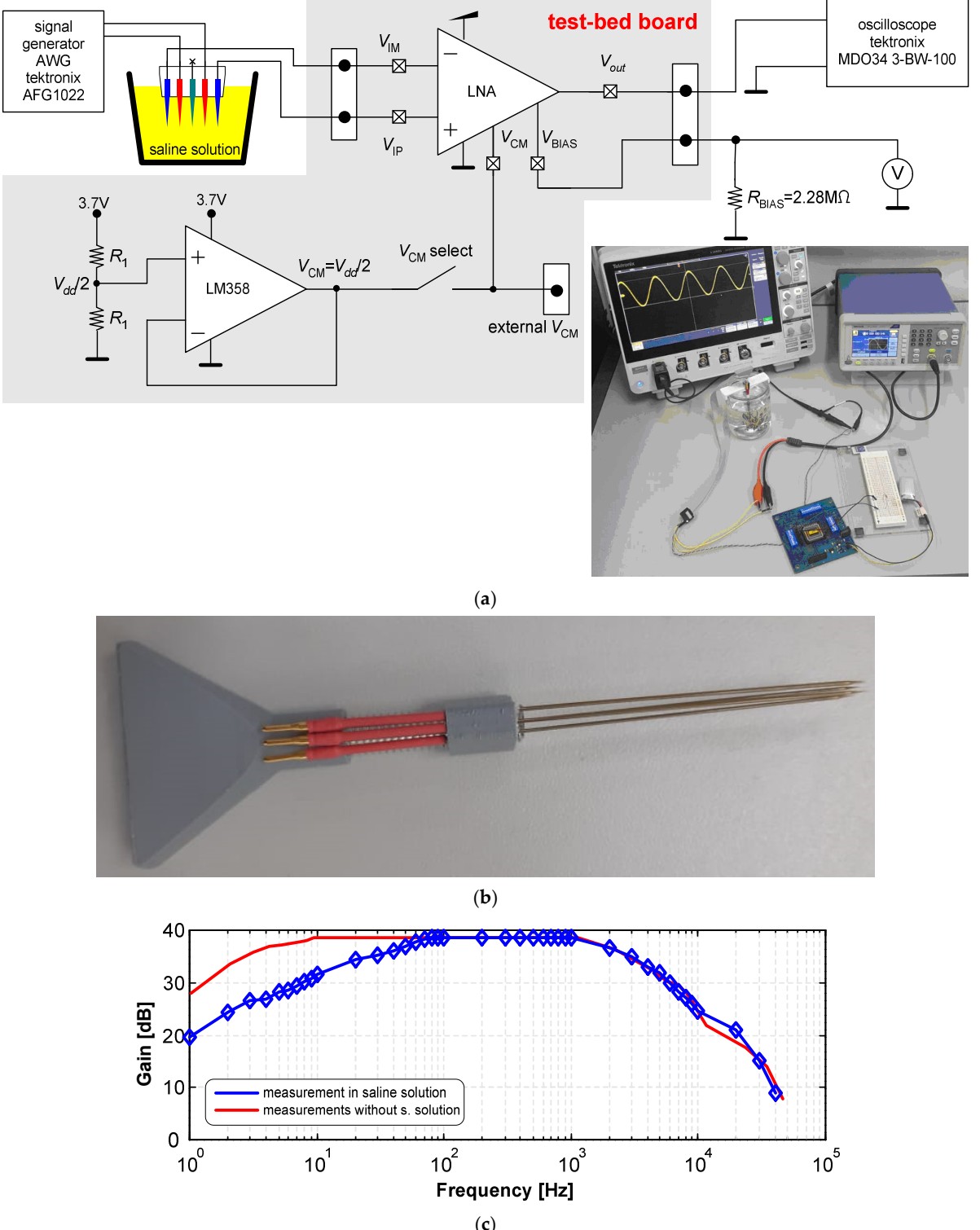

**Figure 14.** (**a**) Schematic of the saline solution setup. The inset shows the setup photograph. (**b**) Photograph of the 3D printed tip with the array of electrodes. (**c**) Measured gain of the LNA.

Figure 14c illustrated the measured gain of the LNA when subjected to these tests. It can be observed that the gain behavior is the same observed in the first tests for frequencies above 100 Hz, and it is slightly lower at frequencies below 100 Hz. The amplitude of the signals injected in the saline solution has 40 $mV_{pp}$ for all frequencies, while the amplitude of the signal sensed in the pair of reading electrodes was also 18 $mV_{pp}$ for all frequencies.

Another set of tests were performed to evaluate the transient responses of the LNA. These tests consisted of the injection of two square waves into the input of the LNA. The amplitudes of these waves were settled to 20 $mV_{pp}$, while their frequencies were settled to 1.15 kHz and 200 Hz. Figure 15a,b shows the signals at the output of the LNA for these frequencies, respectively.

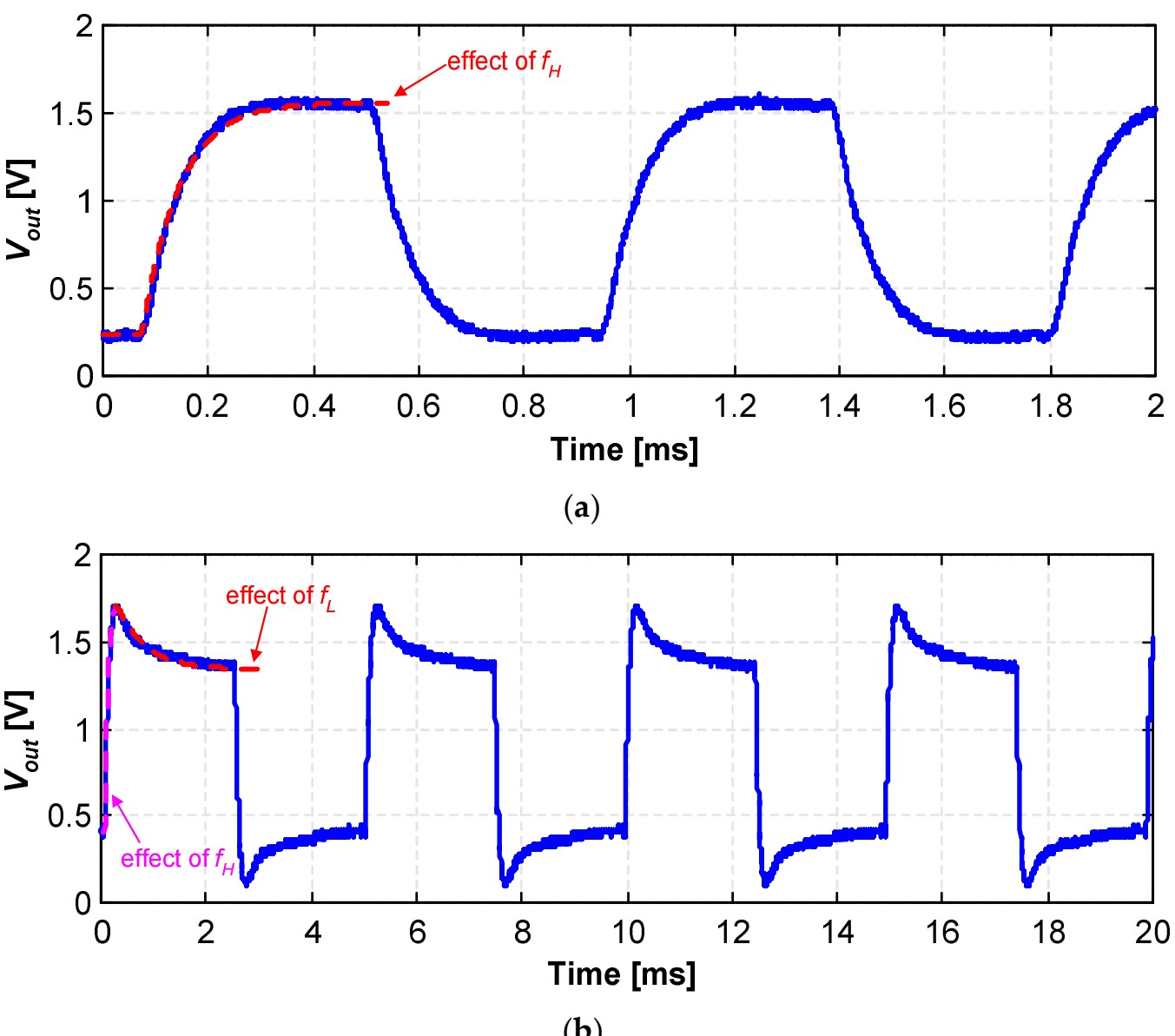

**Figure 15.** LNA response for input waves with square shape, amplitude of 20 mV and frequencies of (**a**) 1.15 kHz and (**b**) 200 Hz.

Figure 15a shows the low-pass effect due to the upper cutoff frequency $f_H$. The frequency of 1.15 kHz is very close to the upper cutoff frequency of the LNA, and it was

selected for this reason. The plot $v_{fitted,H}(t)$, in red, illustrates the effect of the upper cutoff frequency $f_H$ and it was fitted to:

$$v_{fitted,H}(t) = V_{0,H} + A_H \times \left(1 - e^{-\frac{t}{\tau_H}}\right) \tag{15}$$

where $V_{0,H}$ = 0.24 V, $A_H$ = 1.32 V. The time constant $\tau_H$ was calculated in order to $v_{fitted,H}(t)$ agree the best as possible with the rising portion of the output signal. The estimation of this time constant resulted on $\tau_H \approx 69$ µs, meaning an upper cutoff frequency of $f_H = 1/(2\pi\tau_H) \approx 2.31$ kHz, which is practically equal to the $-3$ dB frequency in Figure 11.

Figure 15b shows the high-pass effect due to the lower cutoff frequency $f_L$ of the LNA. The frequency of 200 Hz was selected due to be close to the lower cutoff frequency of the LNA. The plot $v_{fitted,L}(t)$, in red, illustrates the effect of the lower cutoff frequency $f_L$ and was fitted to:

$$v_{fitted,L}(t) = V_{0,L} + A_L \times e^{-\frac{t}{\tau_L}} \tag{16}$$

where $V_{0,L}$ = 1.35 V, $A_L$ = 0.37 V. The time constant $\tau_L$ was also calculated in order to $v_{fitted,L}(t)$ agree the best as possible with the falling portion of the output signal. The estimation of this time constant resulted on $\tau_L \approx 562.9$ µs, meaning a lower cutoff frequency of $f_L = 1/(2\pi\tau_L) \approx 282.7$ kHz. It is interesting to observe that the plot $v'_{fitted,H}(t)$ with the pink trace still illustrates the effect of the upper cutoff frequency $f_H$ and this time was fitted to:

$$v'_{fitted,H}(t) = V'_{0,H} + A'_H \times \left(1 - e^{-\frac{t}{t_H}}\right) \tag{17}$$

where $V'_{0,H} \approx 0.41$ V, $A'_H \approx 1.35$ V. The red and pink plots illustrate the effects of the lower and the higher cutoff frequencies, respectively.

To finish, the common-mode voltage at the output of LNA was measured to be 898 mV ($\approx 0.9$ V or $V_{dd}/2$) in all measurements with the input common-mode $V_{CM}$ equal to 0.9 V.

## 5. Conclusions

This paper presented a low-noise amplifier (LNA) optimized for application on deep-brain stimulation (DBS). This LNA was designed and fabricated in the CMOS 0.18 µm from TSMC. The tests were performed without a buffer in the output of the LNA to achieve the best and complete characterization as possible. The drawback of the absence of a buffer was the rise of parasitic capacitances associated with the conditions of how the tests were performed that were seen by the output of the LNA, more specifically, the contributions of the internal metals of the microdevice, the PADs that connect to the packaging, the ESD protections, the tracks in the test-bed board, the wires and the passive voltage probe of the oscilloscope. Nonetheless, the total contribution was measured and agrees very well with the simulations, meaning that in a final integration this effect is hugely mitigated. Table 2 compares this LNA with few related key works found in the literature [44–53]. It was calculated the figure-of-merit (*FOM*) to better rank and compare this work with the best state of the art [1,44–53] with respect to thermal power-noise trade-off. The noise efficiency factor (*NEF*) was presented in 1987 by Steyaert et al. [57], and since then, it has been widely used and given by:

$$NEF = IRN \times \sqrt{\frac{2I_{total}}{\pi \times U_T \times (4kT) \times BW}} \tag{18}$$

where $I_{total}$ is the total current absorbed by the amplifier stage (this current excludes the amount absorbed by the bias stage), $U_T$ is the thermal voltage given by $kT/q$ ($\approx 26$ mV at the room temperature of 300 K), $k$ is the Boltzmann constant given by $1.38064852 \times 10^{-23}$ m$^2$ kgs$^{-2}$ K$^{-1}$, $T$ the room temperature expressed in Kelvin, and *IRN* [V$_{RMS}$] the input-referred noise. It must be noted that this FOM compares the power-noise trade-off with that of a single ideal bipolar transistor. The lowest the FOM, the better will be the LNA with relation to the global noise performance. The figure-of-merit is defined as the ratio of the $-3$ dB

bandwidth $BW_{kHz}$, expressed in kHz, by the product of the power consumption $P_{\mu W}$ in $\mu W$ with the input-referred noise (*IRN*) in $\mu V_{RMS}$. Table 2 lists and compares this LNA with those found in the literature.

**Table 2.** Comparison of this low-noise amplifier with the state of the art.

| Ref. | CMOS Process | Mid-Band Gain [dB] | Bandwidth [kHz] | Voltage [V] | Power [μW] | Area [mm²] | *IRN* [μV$_{RMS}$] | *FOM* (e.g., the *NEF*) |
|------|--------------|--------------------|-----------------|-------------|------------|------------|---------------------|--------------------------|
| This work | 0.18 μm | 38.6 | 2.3 | 1.8 | 2.8 | 0.035 | 6.2 | 6.19 |
| [1] | 0.13μm | 40.5 | 8.1 | 1 | 12.5 | 0.047 | 3.1 | 4.4 |
| [44] | 28 nm | 51.3 | 3 | 0.5 | 0.9 | N/A | 6.85 | 3.40 |
| [45] | 65 nm | 47.48 | 3 | 0.75 | 6 | N/A | 1.40 | 2.78 |
| [46] | 1.5 μm | 39.5 | 7.2 | ±2.5 | 80 | 0.16 | 2.2 | 3.80 |
| [47] | 0.5 μm | 40.85 | 5.32 | 2.8 | 7.5 | 0.16 | 1.66 | 3.21 |
| [48] | 65 nm | 15 | 10 | 0.5 | 1.1 | 0.004 | 6.5 | 3.71 |
| [49] | 0.18 μm | 50 | 9.2 | 1.2 | 8.6 | 0.05 | 5.6 | 4.90 |
| [50] | 0.13 μm | 40 | 10.5 | 1 | 12.1 | 0.072 | 3.2 | 2.90 |
| [51] | 0.18 μm | 40 | 7.5 | 1.2 | 4.8 | 0.022 | 3.87 | 3.44 |
| [52] | 0.5 μm | 49.26 | 12.9 | 3.3 | 26 | 0.014 | 3.16 | 2.53 |
| [53] | 0.18 μm | 40 | 7.4 | 1 | 3.44 | 0.012 | 4.27 | 3.07 |

To finish, Figure 16 shows a photograph of the fabricated CMOS microdevice, which occupies 1660 μm × 1660 μm of area. Moreover, this figure also makes an emphasis to the LNA presented in this paper and an emphasis to one of the ESD protections for a better illustration and understanding.

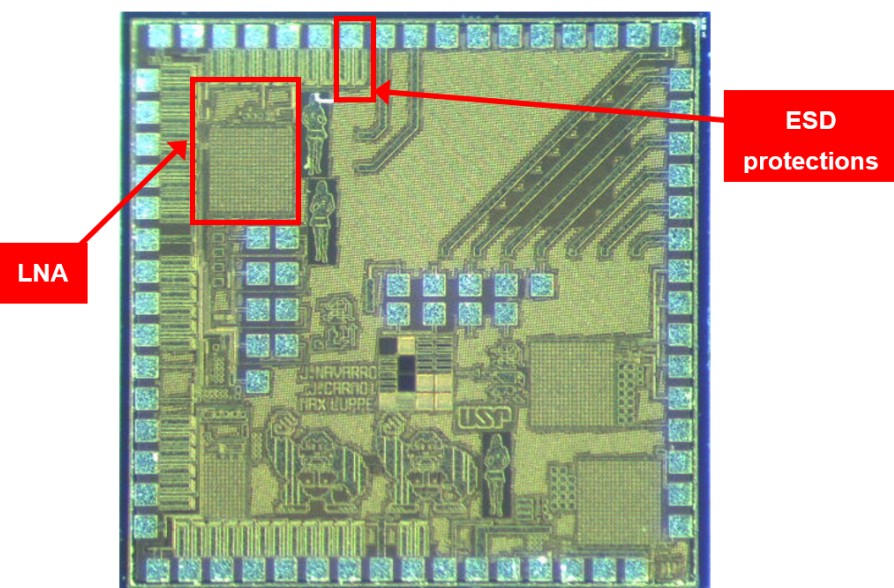

**Figure 16.** Photograph of the fabricated CMOS microdevice (1660 μm × 1660 μm), with emphasis on the LNA presented in this paper and on one of the ESD protections.

**Author Contributions:** Conceptualization, T.M.N., R.H.G. and J.P.P.d.C.; methodology, T.M.N., R.H.G. and J.P.P.d.C.; validation, T.M.N. and R.H.G.; writing—original draft preparation, T.M.N., R.H.G. and J.P.P.d.C.; supervision, E.T.F. and E.C.; project administration, M.L., J.N.S.J., J.P.P.d.C. and M.A.R.; funding acquisition, M.L., J.N.S.J., J.P.P.d.C. and M.A.R. All authors have read and agreed to the published version of the manuscript.

**Funding:** This work was partially supported by the FAPESP agency (Fundação de Amparo à Pesquisa do Estado de São Paulo) through the project with the reference 2019/05248-7. The CMOS microdevice and packaging was offered by IMEC free of any charge, in the scope of IMEC-Brazil program. The CMOS microdevice was fabricated in the scope of IMEC-Brazil program. Tiago Mateus Nordi was sponsored by Federal University of São Carlos (USFCar). Rodrigo Henrique Gounella was supported with a PhD scholarship from CAPES (Coordenação de Aperfeiçoamento de Pessoal de Nível Superior). João Paulo Carmo was support by a PQ scholarship with the reference CNPq 304312/2020-7.

**Acknowledgments:** We would also like to acknowledge to Jacobus Swart from the University of Campinas for his kind and readily available assistance during the whole process and intermediation with IMEC.

**Conflicts of Interest:** The authors declare no conflict of interest.

## Appendix A

*Deduction of the Transfer Function of the LNA*

This deduction takes into account $V_{IM} = V_{in} + V_{ref}$ and $V_{IP} = V_{ref}$, in a similar way as stated by Wattanapanitch et al. [47], where both signals include dc and ac components. According to Figure 3a, the ac output current of the output of the OTA is given by:

$$I_{out} = g_m \times (V^+ - V^-) = -g_m V^- \tag{A1}$$

with $V^+ = 0$ V for ac. This current flows throughout the load impedances $Z_x = (sC_x)^{-1}$ and $Z_2 = R_2/(sR_2C_2 + 1)$ in the feedback path, e.g.,

$$I_{out} = \frac{V_{out}}{Z_x} + \frac{V_{out} - V^-}{Z_2} \tag{A2}$$

The voltage $V^-$ at the inverting input of the OTA is given by:

$$V^- = \frac{Z_2}{Z_1 + Z_2} V_{in} + \frac{Z_1}{Z_1 + Z_2} V_{out} \tag{A3}$$

with $Z_1 = (sC_1)^{-1}$. Thus,

$$-\frac{g_m Z_2}{Z_1 + Z_2} V_{in} - \frac{g_m Z_1}{Z_1 + Z_2} V_{out} = \frac{V_{out}}{Z_x} + \frac{V_{out} - V^-}{Z_2} \tag{A4}$$

replacing the voltage $V^-$ in (A3), the previous equation becomes:

$$-\frac{g_m Z_2}{Z_1 + Z_2} V_{in} - \frac{g_m Z_1}{Z_1 + Z_2} V_{out} = \frac{V_{out}}{Z_x} + \frac{V_{out}}{Z_2} - \frac{V_{in}}{Z_1 + Z_2} - \frac{Z_1}{Z_2(Z_1 + Z_2)} V_{out} \tag{A5}$$

The transfer function can be obtained after few algebraic manipulations, resulting in:

$$H_{LNA} = \frac{Z_x(1 - g_m Z_2)}{Z_x(1 + g_m Z_1) + Z_1 + Z_2} \tag{A6}$$

The first proof is to check the validity of the previous equation of $H_{LNA}$. This is performed by assuming $C_x = 0$ F, thus, $Z_x = \infty$. The transfer function is then:

$$H_{LNA} = \lim_{Z_x \to \infty} \left[ \frac{Z_x(1 - g_m Z_2)}{Z_x(1 + g_m Z_1) + Z_1 + Z_2} \right] = \frac{1 - g_m Z_2}{1 + g_m Z_1} \tag{A7}$$

Since $g_m Z_{1,2} \gg 1$ then:

$$H_{LNA} \approx \frac{-g_m Z_2}{g_m Z_1} = -\frac{Z_2}{Z_1} = -\frac{C_1}{C_2} \tag{A8}$$

The module of $H_{LNA}$ is equal to the mid-band gain of the LNA $A_v = |H_{LNA}| = C_1/C_2$. The transfer function $H_{LNA}$ is negative, but this is not a problem because the input signal $V_{in}$ connects to the inverting input. This analysis was conducted under this assumption to facilitate the demonstration of $H_{LNA}$. For $V_{IM} = V_{ref}$ and $V_{IP} = V_{in} + V_{ref}$, the transfer function would be $H_{LNA} = +C_1/C_2$. The negative signal must be included in the end of this demonstration.

The fully transfer function in terms of all components present in the LNA can be further manipulated to obtain:

$$H_{LNA}(s) = \left(\frac{C_1}{C_2}\right)$$
$$\times \left[\frac{s^2 R_2 C_2 - s(g_m R_2 - 1)}{s^2 R_2 (C_1 + C_x + \frac{C_1 C_x}{C_2}) + s(\frac{C_1}{C_2} + g_m R_2 + \frac{C_x}{C_2}) + \frac{g_m}{C_2}}\right] \tag{A9}$$

The transfer function can be further simplified taking into account that $g_m R_2 \gg 1$ and $g_m R_2 \gg C_i/C_j$ for any combination of $\{C_i, C_j\}$ equal to $\{C_1, C_2, C_x\}$. In this situation:

$$H_{LNA}(s) \approx \left(\frac{C_1}{C_2}\right) \times \left[\frac{(sR_2 C_2 - g_m R_2)s}{s^2 R_2 (C_1 + C_x + \frac{C_1 C_x}{C_2}) + s g_m R_2 + \frac{g_m}{C_2}}\right] \tag{A10}$$

and finally

$$H_{LNA}(s) = \left(\frac{C_1}{C_2}\right)$$
$$\times \left[\left(\frac{R_2 C_2^2}{g_m}\right) \times \frac{\left(s - \frac{g_m}{C_2}\right) \times s}{s^2 \left[\frac{R_2(C_1 C_2 + C_1 C_x + C_2 C_x)}{g_m}\right] + s(R_2 C_2) + 1}\right] \tag{A11}$$

This transfer function can be simplified in terms of the zeros and poles, resulting in:

$$H_{LNA}(s) = \left(\frac{C_1}{C_2}\right) \times \left[\left(\frac{C_2^2}{C_1 C_2 + C_1 C_x + C_2 C_x}\right) \times \frac{\left(s - \frac{g_m}{C_2}\right) \times s}{(s + s_{p1}) \times (s + s_{p2})}\right] \tag{A12}$$

The transfer function contains one zero located in the origin and another located at $f_z = g_m/(2\pi C_2)$ and two poles $s_{p1}$ and $s_{p2}$. The poles of $H_{LNA}(s)$ are given by:

$$\begin{cases} s_{p1} = \frac{g_m C_2}{2(C_1 C_2 + C_1 C_x + C_2 C_x)} \times \left(1 - \sqrt{1 - \frac{4(C_1 C_2 + C_1 C_x + C_2 C_x)}{g_m R_2 C_2^2}}\right) \\ s_{p2} = \frac{g_m C_2}{2(C_1 C_2 + C_1 C_x + C_2 C_x)} \times \left(1 + \sqrt{1 - \frac{4(C_1 C_2 + C_1 C_x + C_2 C_x)}{g_m R_2 C_2^2}}\right) \end{cases} \tag{A13}$$

A further simplification can be made with the poles relation if we consider $g_m R_2 \gg 1$. In this case, we assume that:

$$\frac{4(C_1 C_2 + C_1 C_x + C_2 C_x)}{g_m R_2 C_2^2} \ll 1 \tag{A14}$$

and the poles relations are:

$$\begin{cases} s_{p1} \approx \frac{g_m C_2}{2(C_1 C_2 + C_1 C_x + C_2 C_x)} \times \left(1 - \left(1 + \frac{2(C_1 C_2 + C_1 C_x + C_2 C_x)}{g_m R_2 C_2^2}\right)\right) \\ \quad = \frac{1}{R_2 C_2} \\ s_{p2} \approx \frac{g_m C_2}{2(C_1 C_2 + C_1 C_x + C_2 C_x)} \times (1 + 1) = \frac{g_m C_2}{(C_1 C_2 + C_1 C_x + C_2 C_x)} \\ \quad = \frac{g_m}{(C_1 + (\frac{C_1}{C_2} + 1)C_x)} \end{cases} \tag{A15}$$

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
