# Peer review of "Low-Noise Amplifier for Deep-Brain Stimulation (DBS)"

_electronics, doi:10.3390/electronics11060939_

Round 1

Reviewer 1 Report

This study by T. M. Nordi et al. reports on the design, fabrication, and teste of a low noise amplifier. The work it is interesting, and it deserves to be published. I do not detect major issues with the scientific part of the work, I believe it was well conducted and the amplifier performance is quite interesting and appealing to the readers.

However, there are number of issues related with the quality of the text which are not acceptable for a scientific journal. Namely, the manuscript frequent use of adjectives not really used in scientific writing, poor explanation of the equations which often are mixed with text and unprofessional figures where the experimental set-up is presented as photos of the used equipment.  The ensemble of these aspects gives to the manuscript a careless and very unprofessional aspect.

Before, I can recommend this work for publication, I recommend implementing the following changes in the manuscript:

  • Please review the English in the abstract an in other sections. Some sentences are a bit peculiar.
  • In the introduction section there is a sentence “the recent developments of microelectronics resulted on fantastic and fascinating applications involving the acquisition of biosignals both with wearable and implantable devices” please remove the word “fantastic”.
  • On page 6 it is stated “A clever technique is the use of pseudo-resistors.” Please remove “cleaver”
  • In page 5 and 6 the description of the circuit operation using equations is inserted as text. There are too many equations embed in the text. This looks very strange and difficult to follow. The equations should be written as equations and not inserted in the main text.
  • In figure 4 c the right panel is fuzzy or out of focus. Please correct or remove it.
  • The caption of Figure 6 is written as: “Resistance response of the pseudo-resistors to the voltage input”. Perhaps it is more adequate to say that is “voltage dependence of the pseudo resistance”. I fail to understand why the pseudo resistance is in arbitrary units.
  • Figure 8 is unprofessional for a scientific publication. Please replace it with a proper scientific schematic.
  • Figure 12 (a) is also unprofessional and not appropriate for a scientific journal. Please replace it with a scientific schematic.

Author Response

Please read the first part of the letter addressed to the reviewer 1.

Reviewer 2 Report

Authors presenta a Low-Noise Amplifier (LNA) for the acquisition of biopotentials on DBS. This electronic module was designed in a low-voltage/low-power CMOS process, targeting implantable applications.
The measurement results showed a gain of 38.6dB and a -3dB bandwidth of 2.3kHz. The measurements also showed a power consumption of 2.8uW and an input-referred noise of 3.9uVRMS. The LNA occupies a microdevice area of 122umX283um, promoting the implantation.

In my opinion, the proposed circuit is interesting but description, organization and results presentation have to be improved before acceptance for pubblication. English proof-reading is strongly suggested, some results have to be provided and the work lacks of comparison with the state-of-the-art. 

Particulars of the review are highligheted in the attached file.

Author Response

Please see the comments to reviewer 2 on attached letter.

Round 2

Reviewer 2 Report

The author have addressed my comments.